# An Integrated Solution to FIB-Induced Hydride Artifacts in Pure Zirconium

**DOI:** 10.3390/mi15080999

**Published:** 2024-08-01

**Authors:** Yi Qiao, Zongwei Xu, Shilei Li, Fu Wang, Yubo Huang

**Affiliations:** 1State Key Laboratory for Advanced Metals and Materials, University of Science and Technology Beijing, Beijing 100083, China; lishilei@ustb.edu.cn (S.L.); huangyuboustb@163.com (Y.H.); 2State Key Laboratory of Precision Measuring Technology & Instruments, Laboratory of Micro/Nano Manufacturing Technology, Tianjin University, Tianjin 300072, China; 3018202218@tju.edu.cn

**Keywords:** focused ion beam (FIB), hydrides, zirconium, ion implantation

## Abstract

The preparation method of transmission electron microscopy (TEM) samples for pure zirconium was successfully executed using a focused ion beam (FIB) system. These samples unveiled artifact hydrides induced during the FIB sample preparation process, which resulted from stress damage, ion implantation, and ion irradiation. An innovative solution was proposed to effectively reduce the effect of artifact hydrides for FIB-prepared samples of hydrogen-sensitive materials, such as zirconium alloys. This development lays the groundwork for further research on the micro/nanostructures of zirconium alloys after ion irradiation, thereby facilitating the study of corrosion mechanisms and the prediction of service life for nuclear fuel cladding materials. Furthermore, the solution proposed in this study is also applicable to TEM sample preparation using FIB for other hydrogen-sensitive materials such as titanium, magnesium, and palladium.

## 1. Introduction

Research on cladding materials for nuclear fuel rods, particularly zirconium alloys, is crucial in nuclear power plants due to their demanding service environments. These materials must withstand high-temperature, high-pressure irradiated water externally and neutron irradiation internally [1,2,3,4,5]. Analyzing the micro/nanostructure of zirconium alloys is essential for understanding corrosion mechanisms and developing lifetime prediction models. Ion irradiation is widely accepted as a simulation for neutron irradiation damage because it produces comparable effects without the associated high radiation hazards for researchers [6]. However, ion irradiation penetrates only a few micrometers, making conventional methods like electrochemical polishing and ion thinning inadequate for preparing transmission electron microscopy (TEM) samples. The preferred method for extracting samples from specific areas involves using a focused ion beam (FIB) under online observation.

Despite its advantages, FIB preparation introduces significant issues for hydrogen-sensitive materials such as zirconium alloys, titanium, magnesium, and palladium. Traditional TEM sample preparation using FIB leads to irradiation damage, amorphous damage, and hydride artifacts [7,8,9,10,11,12,13,14,15,16]. These hydride artifacts closely mimic the morphology and phase structure of genuine hydrides, complicating the accurate characterization of the material’s true micro/nanostructures. Consequently, the presence of FIB-induced artifacts can severely hinder our ability to understand the actual conditions and behaviors of the material under study. Thus, it is imperative to develop new methods to mitigate these artifacts and ensure the integrity of TEM sample analyses for hydrogen-sensitive materials.

In this study, we proposed an innovative solution to effectively reduce hydride artifacts in samples prepared by FIB for hydrogen-sensitive materials, including zirconium alloys used in nuclear fuel rod cladding. Our research aims to provide a comprehensive understanding of FIB-induced hydride artifacts in hydrogen-sensitive materials and introduces a new approach for preparing TEM samples of critical materials for nuclear power plants. This solution is particularly relevant for zirconium alloys used in nuclear fuel rod cladding, offering a significant advancement in the accurate characterization of these essential materials.

## 2. Experiment

The material investigated in this study is pure zirconium with 99.8% purity, comprising a single α-Zr phase. The initial hydrogen concentration in the pure zirconium was less than 5 μg/g, indicating that any hydride precipitation observed in the TEM slices likely resulted from subsequent sample processing. To evaluate the generalizability of the confirmation process and the consistency of experimental results, TEM samples were prepared on mechanically polished surfaces using both a Thermo Scientific HX5 FIB-SEM system (Thermo Fisher Scientific, Waltham, MA, USA) and a Zeiss AURIGA FIB-SEM system (CarlZeiss AG, Oberkochen, Germany). Hydride phase characterization was conducted with an FEI Tecnai G2 F30 field TEM (Thermo Fisher Scientific, MA, USA) operating at 300 kV. The TEM sheets were estimated to be approximately 100 nm thick for image acquisition, though the exact thickness was not measured.

### 2.1. TEM Sample Preparation Process

The method of in situ extraction (In situ lift-out) of cross-sectional samples was first implemented by Langford R M et al. [1,2] in 2004 using nano-manipulators and further improved in 2008, where the technique was first implemented as a method of preparing samples perfected by locating a specific region of interest with high accuracy. The TEM sample preparation method in this paper is based on the In situ lift-out method with improvements. In this section, the preparation process and parameters will first be summarized, and the improved method will be described in detail in the following two sections.

The TEM sample preparation process in this study is illustrated in Figure 1:(a)**Platinum Deposition:** A 15 μm × 2 μm × 2 μm layer of platinum is deposited on the sample surface to protect the selected area for preparation. Ion beam-induced deposition uses a focused ion beam to bombard the surface of a sample, inducing the precursor gases of the deposited platinum to decompose and deposit in selected regions of interest on the surface of the sample to form a deposition layer. The ion beam-induced deposition layer has the task of protecting the material surface against scattering ion bombardment in nanofabrication.(b)**Coarse Milling:** The thickness of the sample preparation area is reduced to 2 μm using a large beam (30 kV, 20 nA). The purpose of the rough cut is to initially obtain a thin slice of the sample from the in situ characterization area. The process is to use a larger beam current to process a trapezoidal or rectangular groove above and below the sample deposition protection zone, and the processing frame extends 1~4 μm on each side of the protection zone, with a depth of 5~8 μm and a width of about two times the depth.(c)**Fine Milling:** The thickness of the sample preparation area is further reduced to 1 μm using a smaller beam (30 kV, 2–4 nA). The purpose of the fine cut is to further thin the samples to be extracted in the future and to eliminate the curtain effect caused by the large beam current thinning in the coarse cut stage. At this stage, the thinning will be reciprocated with smaller beam currents on the top and bottom of the sample deposition protection zone in order to minimize stress.(d)**U-Cut:** A U-shaped separation is made between the sample slice and the matrix, leaving a tiny connection. The purpose of the U-cut stage is to pre-separate the matrix from the lamina that will be extracted in the future. At this stage, a smaller beam is used to complete the processing of the asymmetric U-shaped processing frame from left to right on the side of the sample lamina, with a slightly shorter U-shaped edge on the right side to preserve the connection to the matrix.(e)**Lift-Out:** The sample slice is extracted from the matrix using nano-manipulators.(f)**Welding:** The sample slice is welded to a copper holder. Although, as a recognized preparation step, the welding step occurs after the lift-out step, the welding behavior in the typical technological route of the FIB nanofabrication of TEM in situ characterization samples is actually performed two times, once between the sample lamina and the tip of the nano-manipulator W needle and once between the sample lamina and the TEM-specific carrier mesh (holder).

**Figure 1 micromachines-15-00999-f001:**
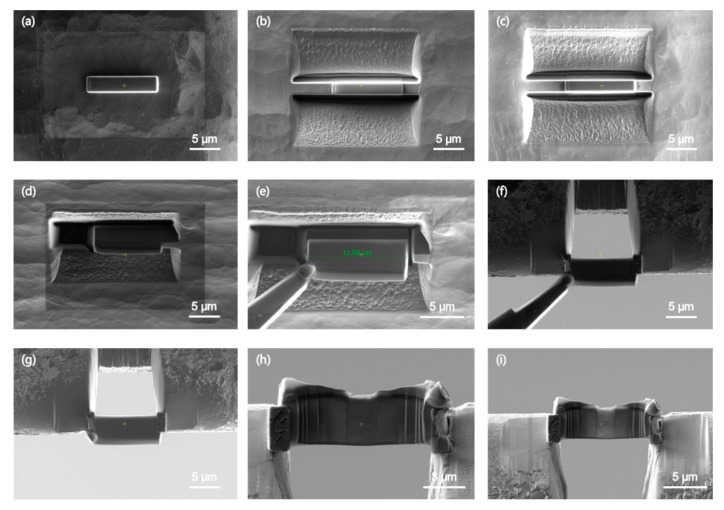
Example of the standard lift-out TEM specimen preparation: (**a**) platinum deposition, (**b**) coarse milling, (**c**) fine milling, (**d**) U-cut, (**e**) lift-out, (**f**) welding, (**g**) separation, (**h**) final thinning, (**i**) cleaning.

The classical welding behavior is Pt deposition welding under an ion beam, which leads to the fracture and dislodgement of the weld due to compressive stress when the nano-manipulator touches the sample lamina before welding and the slight shaking of the sample when cutting the final connection between the sample lamina and the substrate. Thus, the innovative approach of redeposition welding emerged. Redeposition welding is a method that utilizes an auxiliary gas to induce deposited ions and redeposited ions after the material has been bombarded and co-mingled for deposition, which is called the redeposition welding method. The principle is the ion beam processing of the redeposition effect and the slit effect, that is, the high-energy ion beam bombardment of the surface of the material. If, during micro-nano-processing, conducted to form a slit and other small-scale spatial structures, the sputtering ions on the surface of the material cannot escape in a timely manner, they may collide with the slit wall or become adsorbed on the surface of the slit wall when deposited in the small space.

The purpose of the redeposition welding method is to form a more robust welded joint structure and to realize a smooth transfer of an in situ sample. Since redeposition welding is based on the slit effect, the emergence of a slit structure is the basis of the redeposition welding method. Therefore, the selected technology route is as follows: ion beam processing adopts the line milling working mode, and the processing direction is from low-hardness material to high-hardness material or from thin-size material to thick-size material, and high-hardness material or thick-size material is the main one. That is, when the thin sample is connected with the W tip of the nano-manipulator, it is cut from the sample to the W tip; when the thin sample is connected with the TEM special holder, it is cut from the sample to the holder.

(g)**Separation:** The sample slice is separated from the nano-manipulator.(h)**Final Thinning:** The sample slice is thinned further until its thickness is less than 100 nm. The purpose of final thinning is to thin the sample lamina to the thickness required for transmission electron microscopy characterization. In order to avoid the deformation of the sample and reduce stress, double-sided thinning is used, i.e., the sample lamina is thinned alternately on both sides of the sample lamina with a smaller beam current back and forth.(i)**Cleaning:** Amorphous damage on the sample surface is removed using a small beam current at low voltage (e.g., 5 kV, 120 pA).

### 2.2. Point Welding

To reduce stress-induced hydrides, H.H. Shen et al. [11] proposed a final thinning method using a line welding step to minimize sample deformation and associated stress. Building on this method, we propose the point welding graded voltage step-down method to further mitigate stress-induced hydrides.

Figure 2 presents SEM images of different welding methods: (a) the front side of point welding, (b) back side of point welding, (c) front side of line welding, and (d) back side of line welding.

In the proposed point welding method, the left outer lower corner and right outer lower corner of the sample are welded to the copper holder with platinum dots. Conversely, in the original line welding method, the left and right outer sides of the sample are welded to the copper holder with platinum wires. This refinement aims to further reduce stress and resulting hydride artifacts in the samples.

### 2.3. Graded Voltage Thinning

In the final stage of thinning, we implemented a method known as ‘graded voltage thinning’ involving a gradual reduction in both gallium ion beam voltage and current density. To assess the confirmation process’s generalizability and ensure experimental result consistency, transmission electron microscope samples were prepared on mechanically polished surfaces using the Thermo Scientific HX5 FIB-SEM system and the Zeiss AURIGA FIB-SEM system, respectively. Further details on FIB thinning parameters, such as sample slice thickness, accelerating voltage, and beam current, are provided in Table 1 and Table 2.

## 3. Results and Discussion

Figure 3, Figure 4 and Figure 5 depict bright-field transmission electron microscopy (BF-TEM) images showing hydrides in pure zirconium obtained through FIB sampling, employing different welding techniques and thinning processes at 30 kV. The hydride phases induced by FIB appear needle-like, ranging in length from approximately 500 nanometers to over 1 micron in Figure 3, less than 500 nanometers in Figure 4, and in the range of hundreds of nanometers in Figure 5.

Figure 6 presents fast Fourier transformation (FFT) patterns of the zirconium matrix and the hydride phase. The FFT pattern of the zirconium matrix confirms its α-Zr phase with a hexagonal close-packed (HCP) structure in the zone axis of $\left[01\bar{1}0\right]$, while the FFT pattern of the hydride strip confirms a δ-hydride phase with a face-centered cubic (FCC) structure in the zone axis of $\left[0\bar{1}1\right]$. Figure 7 shows BF-TEM images of hydrides in pure zirconium obtained through FIB sampling with point welding, line welding, and graded voltage thinning processes conducted by AURIGA.

From Figure 3, Figure 4, Figure 5, Figure 6 and Figure 7, it is evident that the hydrides introduced from the zirconium matrix (HCP) during FIB sample preparation appeared as δ-hydride structures with a needle-like FCC morphology. The hydrides resulting from FIB preparation in welding processes exhibited varying thicknesses: those from point welding were finer compared to the thicker hydrides from other methods. While graded voltage thinning reduced hydride presence compared to direct thinning at 30 kV, the overall size did not show significant change. Various FIB-SEM models employing graded thinning processes demonstrated consistent results.

### 3.1. Mechanism of Hydride Introduction

The completion of micro/nano characterization sample preparation via FIB milling occurs within a high-vacuum environment (typically below 10^−5^ mbar), thereby minimizing the occurrence of hydride artifacts. Wunk et al. have presented a compelling rationale for hydrogen generation, attributing it to the desorption reaction of organic metal precursors from the gas injection system (GIS) during electron or ion irradiation [17]. The reaction is depicted as follows:(1)$${MeCpPt}^{IV}Me_{3}(Precursor)+e^{-}\to {PtC}_{8}(Adsorbed\;PtC_{x}\;film+H_{2}(gas)↑+\;{CH}_{4}(gas)↑)$$

The use of Pt organometallic compounds (specifically methylcyclopentadienyl platinum) is incorporated extensively in the FIB sample preparation process, spanning the entire TEM sample preparation workflow [18]. This includes safeguarding the sample surface, extracting the sample, and welding the sample, as illustrated in Figure 8.

There are two primary mechanisms responsible for the introduction of hydride artifacts: (1) The stress mechanism triggers the introduction of hydride artifacts. Shen et al. [11] observed that during the final thinning process using a focused ion beam (FIB), stress relaxation resulted in the deformation of the final transmission electron microscopy (TEM) sample, creating localized strain fields within the sample. These strain fields provide the necessary stress conditions for the nucleation of hydride artifacts. (2) Another significant factor influencing the introduction of hydride artifacts is the surface temperature of the sample. The thermal effect generated by FIB cutting accelerates the introduction and diffusion of hydrogen, thereby promoting the nucleation and growth of hydride artifacts [1]. It is noted that the solubility of hydrogen in zirconium rapidly increases with rising temperatures.

### 3.2. Solution to FIB-Induced Artifact Hydrides

Presently, derived from the origin of hydride artifacts, there exist five methodologies aimed at mitigating these artifacts: (1) decreasing the hydrogen source, such as regulating the total amount of Pt deposition prudently and exploring alternative Pt sources; (2) augmenting the vacuum level, such as substituting conventional ion pumps with titanium sublimation pumps; (3) employing a cold stage to reduce hydrogen solubility; (4) limiting sample deformation to lessen stress; and (5) mitigating the thermal impacts of FIB processing.

The initial three approaches focus on hardware enhancements, contrasting with the latter two which involve technical investigations. Currently, organic compounds serve as the primary sources of various gases, with hydrogen being a ubiquitous component across these sources. Additionally, assembling titanium sublimation pumps presents challenges, prompting researchers to favor cold stages. Chang et al. [9] conducted a comparative study using samples prepared via three-dimensional atomic probes (3DAPs) and TEM under low-temperature conditions (−120 °C) versus conventional room-temperature samples. Their findings demonstrate that ultra-low-temperature cutting effectively reduces artifact hydride formation [9]. However, due to its high cost, cold stages are installed in only a limited number of FIB devices. Consequently, most researchers are directing their efforts towards exploring and refining technical methodologies.

Shen et al. [11] employed a stepwise thinning method with line welding to produce non-deformed TEM samples, minimizing the introduction of stress-induced artifact hydrides. Additionally, several researchers have explored a 5 kV voltage cleaning technique to eliminate amorphous and artifact hydrides from the final thinned samples. However, while effective for preparing TEM samples of amorphous materials, this method has limited efficacy for artifact hydrides.

Building on this background, our research team prioritized reducing hydride introduction during sample preparation at room temperature. Our technical approach involves (1) mitigating stress and (2) minimizing thermal effects.

Below is an innovative solution proposed to minimize the artifact hydrides:(1)Point welding

To alleviate stress, we employed the line welding progressive thinning method proposed by Shen et al. [11] in this study. This method was successfully utilized to procure undistorted central TEM specimens. However, despite these achievements, a notable presence of artifact hydrides persisted, as evidenced in Figure 3 and Figure 7. This issue is attributed to the comprehensive fixation caused by the line welding on both sides of the sample. The initial extensive welding induced substantial thermal effects, leading to contraction during cooling. The fixed welding points restricted inward contraction, resulting in tensile stresses. Additionally, the subsequent bidirectional thinning process further exacerbated stress due to the constrained sample sides, which hindered stress release and concentrated stress within the sample. Consequently, although the sample remained deformation-free, a significant stress concentration occurred, contributing to the formation of hydride artifacts.

Based on the above analysis, point welding was proposed, illustrated in Figure 2a,b. In point welding, the lift-out lamella was fused into the Cu holder with contact points on both the left and right sides. The primary advantage lies in reducing Pt deposition during sample preparation due to the smaller contact area between the lamella and Cu grid. This method effectively minimizes hydrogen resources in the chamber. Additionally, point welding imposes less stress on the TEM lamella, releasing thermal-induced stress during thinning. The hydride artifact in the TEM lamella prepared using point welding was significantly reduced compared to line welding, as depicted in Figure 3. Thus, employing point welding for lamella fusion proves to be an effective strategy in minimizing hydride artifacts.

(2)Graded voltage thinning

In recent years, with the extensive and in-depth application of a focused ion beam in nanofabrication, the study of amorphous damage has gradually received the attention of scientific workers. Since focused ion beams were firstly applied to the semiconductor industry, the study of amorphous damage has been mainly based on Si materials, and it has been found that the accelerating voltage of a focused ion beam shows a close correlation with the thickness of the amorphous layer of the sample [19,20,21,22,23,24,25,26]. The dose of the ion beam current, the processing time, the crystal orientation of the sample, the thickness (the effect is obviously intensified after reaching less than a hundred nanometers), and the atomic number of the material affect the depth of the amorphous damage. When the atomic number of the material is low, such as in C, Al, Si, and other semiconductors and lightweight metals that are amorphous-sensitive materials, when accelerating the voltage up to 30 kV, increasing amorphous damage up to 20 nm, and reducing the cutting beam voltage down to 2–5 kV, the thickness of the amorphous layer can be reduced to 1–5 nm [19,20,21,22,23], so low-voltage cleaning is indispensable. It is usually believed that the thickness of the amorphous damage layer will be reduced when the material has a higher atomic number [24,25,26], but the amorphous damage of Zr alloys is not clear, so this paper made an attempt to use graded voltage thinning of 30–15–10 kV, and SRIM simulated the thickness of the amorphous damage under different voltages. In order to maximize the elimination of amorphous damage, the sample surface was cleaned at the final stage of sample preparation using a 5 kV voltage beam current, and the diffraction was carried out by low-voltage cleaning, and the diffraction results showed that the diffraction spots were clear without amorphous rings.

During the final stages of thinning, we implemented a technique known as ‘gradual voltage thinning’. This involved a progressive reduction in both the voltage and current density of the Ga ion beam. A simulation using SRIM was conducted to analyze the distribution of Ga ions and dislocation defects during ion beam milling at various energy levels in zirconium. The simulations revealed that at energy configurations of 10 KeV, 15 KeV, and 30 KeV, the maximum penetration depths of ion injection were measured at 20 nm, 23 nm, and 50 nm, respectively, with penetration depth increasing alongside energy levels (Figure 9a–c). Ga ions injected at different energies exhibited a roughly Gaussian distribution within Zr, peaking at depths of 3 nm, 5 nm, and 6 nm beneath the surface of pure Zr, with corresponding peak distribution densities of 46 × 10^4^ ATOMS/cm^2^, 37 × 10^4^ ATOMS/cm^2^, and 24 × 10^4^ ATOMS/cm^2^.

An analysis of the damage distribution map indicated that Ga ions injected into the Zr surface underwent cascade collisions, resulting in a decreasing number of vacancies with increasing depth (Figure 9d–f). The cumulative number of displacements caused by a single ion beam at different depths was found to be 74, 105, and 202, respectively. Vacancies generated by different energy injections exhibited maximum impact depths of 11 nm, 15 nm, and 31 nm, respectively, with a concentration of dislocation defects predominantly observed at depths exceeding 10 nm beneath the surface of pure Zr.

## 4. Conclusions

This paper introduces a TEM sample preparation technique tailored for hydrogen-sensitive materials such as a zirconium alloy, titanium, magnesium, and palladium using a focused ion beam (FIB) system. The findings demonstrate that employing point welding and graded voltage thinning methods effectively mitigates artifact hydrides compared to conventional zirconium alloy sample preparation methods for nuclear materials. Point welding and progressive thinning are recommended during sample preparation, significantly aiding in capturing the true micro/nanostructures of the material and facilitating the acquisition of TEM samples of the zirconium alloy. Furthermore, this approach minimizes the amorphous damage layer induced by Ga ions. These advancements are crucial for advancing research on irradiation damage and predicting the service life of nuclear fuel cladding materials.

## Figures and Tables

**Figure 2 micromachines-15-00999-f002:**
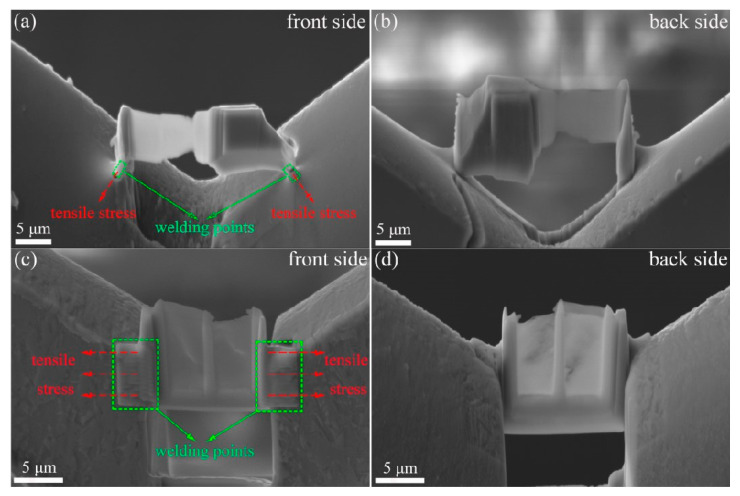
SEM images of different welding methods: (**a**) front-side image of point welding, (**b**) back-side image of point welding, (**c**) front-side image of line welding, (**d**) back-side image of line welding.

**Figure 3 micromachines-15-00999-f003:**
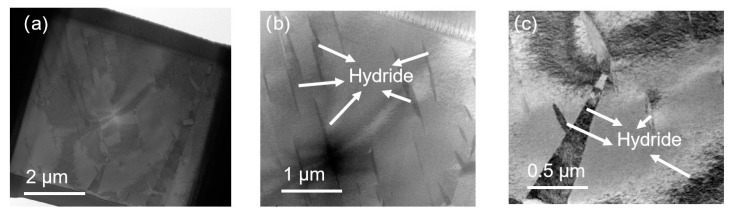
BF-TEM images of hydride in pure zirconium based on FIB sampling with line welding and 30 kV thinning processes by HX5. (**a**–**c**) are sequentially enlarged bright-field morphology images of hydrides.

**Figure 4 micromachines-15-00999-f004:**
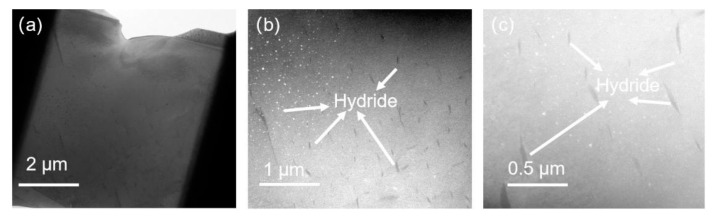
BF-TEM images of hydride in pure zirconium based on FIB sampling with point welding and 30 kV thinning processes by HX5. (**a**–**c**) are sequentially enlarged bright-field morphology images of hydrides.

**Figure 5 micromachines-15-00999-f005:**
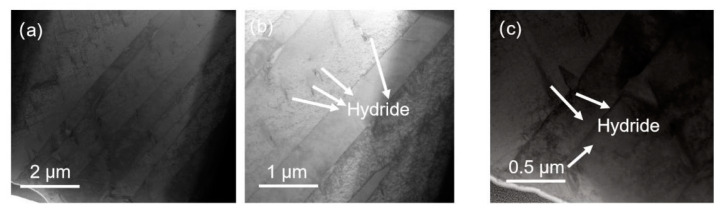
BF-TEM images of hydride in pure zirconium based on FIB sampling with point welding and graded voltage thinning processes by HX5. (**a**–**c**) are sequentially enlarged bright-field morphology images of hydrides.

**Figure 6 micromachines-15-00999-f006:**
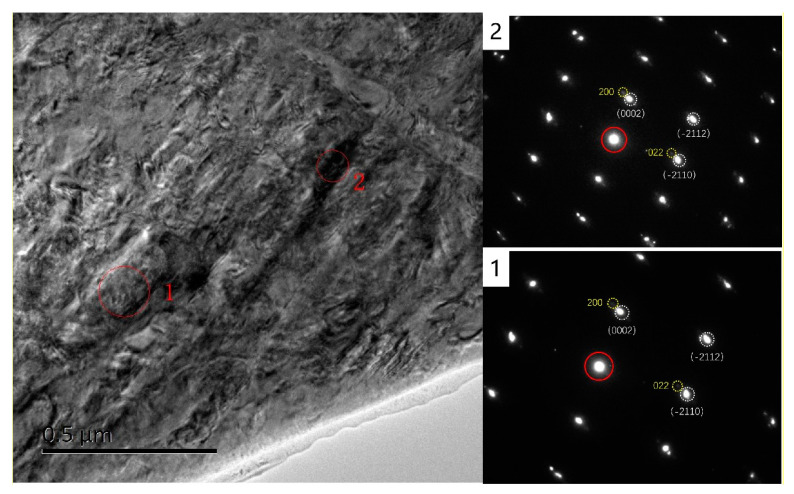
TEM observation and diffraction analysis of hydride artifacts with point welding and graded voltage thinning by HX5. 1 and 2 are selected electron diffraction of hydrides artifacts at different locations.

**Figure 7 micromachines-15-00999-f007:**
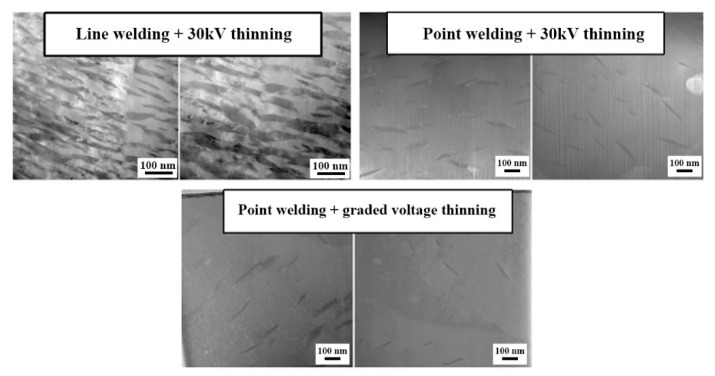
BF-TEM images of hydride in pure zirconium based on FIB sampling with point welding, line welding, and graded voltage thinning processes by AURIGA.

**Figure 8 micromachines-15-00999-f008:**
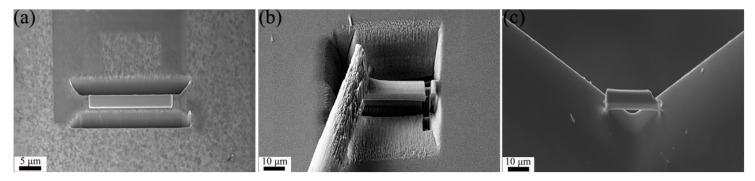
The application of Pt in FIB sample preparation: (**a**) Pt deposition of the protective layer, (**b**) lift-out, (**c**) welding to a copper holder.

**Figure 9 micromachines-15-00999-f009:**
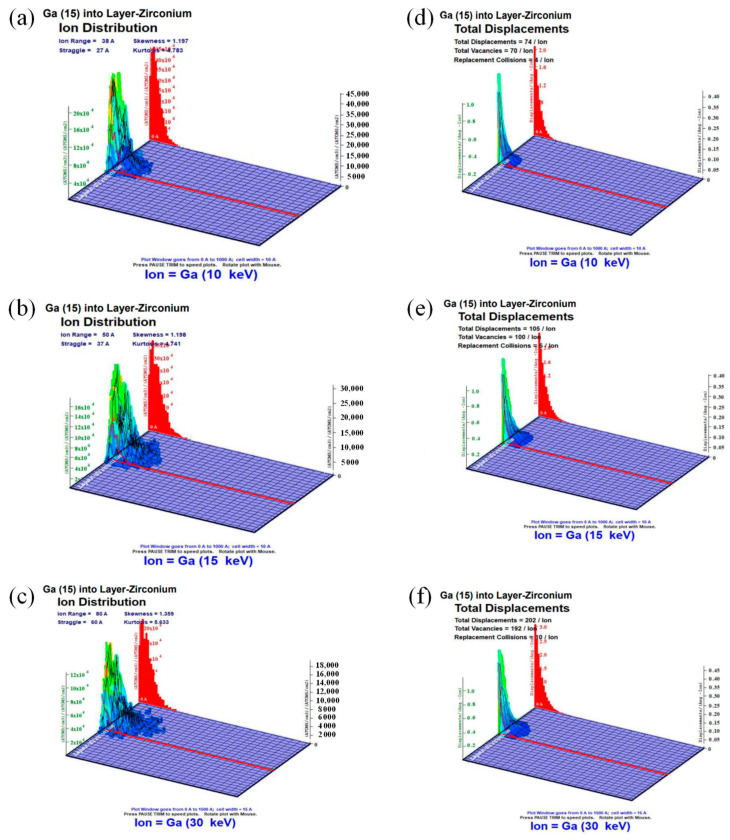
SRIM simulation of 3D ion distributions (**a**–**c**) and total displacements (**d**–**f**) of Ga ions injected into zirconium with 10 KeV, 15 KeV, and 30 KeV energy configurations, respectively.

**Table 1 micromachines-15-00999-t001:** Sample thickness and ion beam processing parameters by HX5.

Sample Thickness H/nm	Ion Beam Voltage U/kV	Ion Beam Current I/pA
1000	30	750
200	16	230–120
100	12	92
50	8	61

**Table 2 micromachines-15-00999-t002:** Sample thickness and ion beam processing parameters by AURIGA.

Sample Thickness H/nm	Ion Beam Voltage U/kV	Ion Beam Current I/pA
1000	30	240–120
200	15	250
100	15	120
50	10	50

## Data Availability

The original contributions presented in the study are included in the article material, and further inquiries can be directed to the corresponding authors.

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
