# Peer review of "An Integrated Solution to FIB-Induced Hydride Artifacts in Pure Zirconium"

_micromachines, 2024, doi:10.3390/mi15080999_

Round 1
Reviewer 1 Report
Comments and Suggestions for Authors
This work proposes an innovative solution to effectively reduce the introduction of hydride artefacts in samples prepared by FIB for hydrogen-sensitive materials. The manuscript can be accepted after minor revisions. Below are my comments:
1- Please enrich the introduction by adding a paragraph to highlight the current knowledge and the novelty of this manuscript. Th introduction is short and needs to be improved.
2- Figure 4, the font on the figures needs to be increase in size for clarity.
3- More references need to be cited, with a particular focus on the recent publications.
Comments on the Quality of English Language
This work proposes an innovative solution to effectively reduce the introduction of hydride artefacts in samples prepared by FIB for hydrogen-sensitive materials. The manuscript can be accepted after minor revisions. Below are my comments:
1- Please enrich the introduction by adding a paragraph to highlight the current knowledge and the novelty of this manuscript. Th introduction is short and needs to be improved.
2- Figure 4, the font on the figures needs to be increase in size for clarity.
3- More references need to be cited, with a particular focus on the recent publications.
Author Response
Dear Reviewers:
Thank you for your comments concerning our manuscript entitled “An integrated solution to FIB induced hydride artefacts in pure zirconium” (micromachines-2990687). Those comments and suggestions are very helpful for improving our paper. We have studied the comments carefully and have made corresponding revisions or corrections. The responses to the reviewer’s comments are listed as follows point by point:
Responses to the Reviewer 1:
Reviewer #1:
1-Please enrich the introduction by adding a paragraph to highlight the current knowledge and the novelty of this manuscript. The introduction is short and needs to be improved.
Thank you for your comments. We have added the advantages and innovations of this work in introduction.
“Research on cladding materials for nuclear fuel rods, particularly zirconium alloys, is crucial in nuclear power plants due to their demanding service environments. These materials must withstand high-temperature, high-pressure irradiated water externally and neutron irradiation internally [1-5]. Analyzing the micro/nanostructure of zirconium alloys is essential for understanding corrosion mechanisms and developing lifetime pre-diction models. Ion irradiation is widely accepted as a simulation for neutron irradiation damage because it produces comparable effects without the associated high radiation hazards for researchers [6]. However, ion irradiation penetrates only a few micrometers, making conventional methods like electrochemical polishing and ion thinning inadequate for preparing high-resolution transmission electron microscopy (HRTEM) samples. The preferred method for extracting samples from specific areas involves using a focused ion beam (FIB) under online observation.
Despite its advantages, FIB preparation introduces significant issues for hydrogen-sensitive materials such as zirconium alloys, titanium, magnesium, and palladium. Traditional TEM sample preparation using FIB leads to irradiation damage, amorphous damage, and hydride artefacts [7-14]. These hydride artefacts closely mimic the morphology and phase structure of genuine hydrides, complicating the accurate characterization of the material's true micro/nanostructures. Consequently, the presence of FIB-induced artefacts can severely hinder our ability to understand the actual conditions and behaviors of the material under study. Thus, it is imperative to develop new methods to mitigate these artefacts and ensure the integrity of TEM sample analyses for hydrogen-sensitive materials.
In this study, we proposed an innovative solution to effectively reduce hydride artefacts in samples prepared by FIB for hydrogen-sensitive materials, including zirconium alloys used in nuclear fuel rod cladding. Our research aims to provide a comprehensive understanding of FIB-induced hydride artefacts in hydrogen-sensitive materials and introduces a new approach for preparing TEM samples of critical materials for nuclear power plants. This solution is particularly relevant for zirconium alloys used in nuclear fuel rod cladding, offering a significant advancement in the accurate characterization of these essential materials.”
2-Figure 4, the font on the figures needs to be increase in size for clarity.
Thank you for your comments. We have corrected this error and added some figures to show more details and content of the experiment.
3-More references need to be cited, with a particular focus on the recent publications.
Thank you for your comments. We have added the some recent publications about this work.
Reviewer 2 Report
Comments and Suggestions for Authors
In order to to effectively reduce the effect of artefact hydrides for FIB-prepared samples of hydrogen-sensitive materials like zirconium alloy, a series of tests were carried out in this paper. However, the following problems need to be solved before publication:
1. English corrections are needed as they are many misspelling and grammatical errors.
2. The abstract needs to succinctly state what problem, what solution, what research program, what work has been carried out and what results have been achieved for this paper. Please revise and refine it carefully.
3. It is strongly recommended to present the schematic diagram of the sample processing.
4. It is suggested to add the corresponding EDS mappings In Figure 3, which is important for confirming the distribution of hydride artefacts.
5. Figure 2 and 4 are so blurry that it is impossible to accurately capture important information, please provide a clearer picture if possible.
6. As stated in the Introduction, the traditional TEM sample preparation method using FIB will introduce the irradiation damage, amorphous damage. Then, how about the present sample preparation method with regard to the irradiation damage and amorphous damage?
7. Conclusion: This paper does not provide a point-by-point account of the conclusions, so if possible please refine the conclusions section and break it down into points. Moreover, it is necessary to make a general summary at the beginning of the conclusion.
Comments on the Quality of English Language
English corrections are needed as they are many misspelling and grammatical errors.
Author Response
Dear Reviewers:
Thank you for your comments concerning our manuscript entitled “An integrated solution to FIB induced hydride artefacts in pure zirconium” (micromachines-2990687). Those comments and suggestions are very helpful for improving our paper. We have studied the comments carefully and have made corresponding revisions or corrections. The responses to the reviewer’s comments are listed as follows point by point:
Responses to the Reviewer 2:
Reviewer #2:
- English corrections are needed as they are many misspelling and grammatical errors.
Thank you for your comments. We have corrected these errors.
- The abstract needs to succinctly state what problem, what solution, what research program, what work has been carried out and what results have been achieved for this paper. Please revise and refine it carefully.
Thank you for your comments. We have revised and rewritten the abstract carefully.
“The preparation method of transmission electron microscopy (TEM) samples for pure zirconium was successfully executed using a focused ion beam (FIB) system. These samples unveiled artefact hydrides induced during the FIB sample preparation process, which are resulting from stress damage, ion implantation and ion irradiation. An innovative solution was proposed to effectively reduce the effect of artefact hydrides for FIB-prepared samples of hydrogen-sensitive materials, such as zirconium alloy. This development lays the groundwork for further research on the micro/nanostructures of zirconium alloy after ion irradiation, thereby facilitating the study of corrosion mechanisms and the prediction of service life for nuclear fuel cladding materials. Furthermore, the proposed solution proposed in this study is also applicable to the TEM sample preparation using FIB for other hydrogen-sensitive materials, such as titanium, magnesium, and palladium.”
3.It is strongly recommended to present the schematic diagram of the sample processing.
Thank you for your comments. We have added figures of this (Figure 1 and Figure 2).
4.It is suggested to add the corresponding EDS mappings In Figure 3, which is important for confirming the distribution of hydride artefacts.
Thank you for your comments. We have added diffraction information from TEM to prove hydride artefacts exists. Generally, the EDS in the TEM uses a Be detector, which can't detect hydrogen atoms.
- Figure 2 and 4 are so blurry that it is impossible to accurately capture important information, please provide a clearer picture if possible.
Thank you for your comments. We have corrected this error.
- As stated in the Introduction, the traditional TEM sample preparation method using FIB will introduce the irradiation damage, amorphous damage. Then, how about the present sample preparation method with regard to the irradiation damage and amorphous damage?
Thank you for your comments. The sample preparation method of this work can substantially weaken irradiation damage and amorphous damage by graded thinning and milling.
- Conclusion: This paper does not provide a point-by-point account of the conclusions, so if possible please refine the conclusions section and break it down into points. Moreover, it is necessary to make a general summary at the beginning of the conclusion.
Thank you for your comments. We have revised and rewritten the conclusion carefully.
“This paper introduces a TEM sample preparation technique tailored for hydro-gen-sensitive materials such as zirconium alloy, titanium, magnesium, and palladium using a focused ion beam (FIB) system. The findings demonstrate that employing point-welding and graded voltage thinning methods effectively mitigates artefact hydrides compared to conventional zirconium alloy sample preparation methods for nuclear materials. Point-welding and progressive thinning are recommended during sample preparation, significantly aiding in capturing the true micro/nanostructures of the material and facilitating the acquisition of TEM samples of zirconium alloy. Furthermore, this approach minimizes the amorphous damage layer induced by Ga ions. These advancements are crucial for advancing research on irradiation damage and predicting the service life of nuclear fuel cladding materials.”
Round 2
Reviewer 2 Report
Comments and Suggestions for Authors
Most comments are well addressed, and the manuscript was improved substantially after revision. However, the author stated that the sample preparation method of this work can substantially weaken irradiation damage and amorphous damage by graded thinning and milling, the related data or references is needed to support this statement.
Comments on the Quality of English LanguageMinor editing of English language required
Author Response
Dear Reviewers:
Thank you for your comments concerning our manuscript entitled “An integrated solution to FIB induced hydride artefacts in pure zirconium” (micromachines-2990687). Those comments and suggestions are very helpful for improving our paper. We have studied the comments carefully and have made corresponding revisions or corrections. The revised parts are highlighted in red in the revised manuscript. The responses to the reviewer’s comments are listed as follows point by point:
Responses to the Reviewer :
Most comments are well addressed, and the manuscript was improved substantially after revision. However, the author stated that the sample preparation method of this work can substantially weaken irradiation damage and amorphous damage by graded thinning and milling, the related data or references is needed to support this statement.
Thank you for your comments. We have added the description and references in section 3.2.
“In recent years, with the extensive and in-depth application of focused ion beam in nanofabrication, the study of amorphous damage has gradually received the atten-tion of scientific workers. Since the focused ion beam is firstly applied to the semicon-ductor industry, the study of amorphous damage is mainly based on Si materials, and it is found that the accelerating voltage of the focused ion beam shows a close correla-tion with the thickness of the amorphous layer of the sample [19-26]. The dose of the ion beam current, processing time, the crystal orientation of the sample, the thickness (the effect is obviously intensified after reaching less than a hundred nanometers) and the atomic number of the material will affect the depth of the amorphous damage. When the atomic number of the material is low, such as C, Al, Si and other semicon-ductors and lightweight metals are amorphous sensitive materials, when the acceler-ating voltage up to 30 kV amorphous damage up to 20 nm, cutting beam voltage down to 2 kV-5 kV, the thickness of the amorphous layer can be reduced to 1-5 nm [19-23], so the low-voltage cleaning is indispensable. It is usually believed that the thickness of the amorphous damage layer will be reduced when the material with higher atomic num-ber [24-26], but the amorphous damage of Zr alloys is not clear, so this paper made an attempt of graded voltage thinning of 30 kV-15 kV-10 kV, and SRIM simulated the thickness of the amorphous damage under different voltages, and in order to eliminate the amorphous damage to the greatest extent possible, in the final stage of the sample making, the sample surface was cleaned by using a beam current of 5 kV voltage, and the diffraction was carried out by a low voltage cleaning. In order to maximize the elimination of amorphous damage, the sample surface was cleaned at the final stage of sample preparation using a 5 kV voltage beam current, and the diffraction results showed that the diffraction spots were clear without amorphous rings.”